# Therapeutic Approaches for Advanced Basal Cell Carcinoma: A Comprehensive Review

**DOI:** 10.3390/cancers17010068

**Published:** 2024-12-29

**Authors:** Magdalena Hoellwerth, Matthias Brandlmaier, Peter Koelblinger

**Affiliations:** Department of Dermatology and Allergology, Paracelsus Medical University, Muellner Hauptstraße 48, 5020 Salzburg, Austria; m.hoellwerth@salk.at (M.H.); m.brandlmaier@salk.at (M.B.)

**Keywords:** hedgehog inhibitors, advanced BCC, sonidegib, vismodegib, PD-1, immune-checkpoint inhibitors, non-melanoma skin cancer

## Abstract

This review provides an overview of therapeutic approaches for advanced basal cell carcinoma (BCC), focusing on both locally advanced (laBCC) and metastatic (mBCC) disease. It entails a general overview of advanced BCC management and outlines approved systemic therapies, including hedgehog inhibitors (HHI) and immunotherapy. The review also explores ongoing research regarding emerging treatment strategies.

## 1. Introduction

Eight out of ten skin tumors are basal cell carcinomas (BCC) [1]. Increased life expectancy, ultraviolet light-exposure, genetic predisposition, male sex, immunosuppression, and other environmental factors contribute to an annual increase of 1% in incidence of BCC, with a lifetime risk of approximately 30% amongst fair-skinned individuals [2,3,4,5]. The true incidence of BCC is difficult to determine, as incomplete or absent reporting of BCC cases in cancer registries may contribute to an underestimation of BCC frequency, resulting in incoherent data among nations [5,6]. Hence, annual incidence rates for BCC vary widely. Estimations range from 76 to 165 new cases per 100,000 inhabitants in European countries [7,8]. In general, the likelihood of developing BCC increases with age, with a median age of onset of 68 years [9]. In addition to the histopathologic classification of the primary tumor, BCC is often classified as localized, locally advanced (laBCC), or metastatic (mBCC) [10,11,12,13]. Recently, an alternative staging system, distinguishing common ‘easy-to-treat’ BCC and different types of ‘difficult-to-treat’ BCC, has been proposed [14,15]. While localized BCC can be treated with potentially curative surgery or topical therapy, the management of laBCC and mBCC requires a multidisciplinary approach, eventually frequently including systemic treatment, typically after local surgical and radiotherapeutic measures have failed. While chemotherapy, despite limited efficacy, was the treatment of choice for extensive disease just a few decades ago, the approval of hedgehog inhibitors, namely vismodegib and sonidegib, have revolutionized systemic therapy in advanced stages [10,16,17]. This review provides an overview of established and recently approved systemic treatments for advanced basal cell carcinoma, comparing available pivotal trial data, and concluding with an outlook on potential future treatment strategies.

## 2. Molecular Pathogenesis of BCC: The Sonic Hedgehog Pathway

In the 1990s, genetic analyses of Gorlin Syndrome families found an allelic loss on chromosome 9q22, linked to the human homologue of Patched (PTCH1), showing germline mutations in one allele of patients. Affected individuals were observed to develop multiple basal cell carcinomas in young adulthood, indicating a role of above genetic aberration in these tumors [18]. The tumor suppressor-gene PTCH1 encodes a transmembrane receptor protein for hedgehog family signaling molecules such as sonic hedgehog (SHH). SHH signaling regulates cell growth and differentiation during embryogenesis, but is suppressed in adults through PTCH1 in the absence of SHH [19,20]. Activation of the SHH signaling pathway occurs through the binding of SHH to the PTCH1 receptor, which relieves the inhibitory effect of PTCH1, hence disinhibiting a surface membrane protein named “Smoothened” (SMO). SMO is a G-protein linked transmembrane receptor and promotes signal transduction through Glioma associated oncogene 1 (GLI1) activation. Consequently, this results in the nucleus-bound upregulation of oncogenes. Particularly mutated PTCH1 may further disinhibit SMO until SHH-independent pathway activation occurs, eventually resulting in cell cycle disruptions, uncontrolled proliferation and ultimately tumor formation (Figure 1) [19]. Genetic and molecular analyses of sporadic BCCs detected PTCH mutations in 70% and SMO mutations in 10% of tumors [21]. Furthermore, other UV-induced somatic mutations, and the inhibition of the tumor-suppressor-gene Tp53, contribute to BCC development [22,23].

## 3. Clinical Presentation

The clinical presentation of BCC is heterogeneous, ranging from small superficial lesions to large tumors invading adjacent tissue [24]. Chronically sun-exposed areas of the skin are most frequently affected. Eighty percent of BCCs are located on the head and neck, 13.4% on the trunk, 3.8% on the upper extremity, 1.5% on the lower extremity, and 0.2% in the genital area [25].

BCCs are considered slow-growing tumors with a low tendency to metastasize. They are prone to ulceration and destructive growth, potentially leading to cosmetic disfigurement [26]. With an increasing emphasis on preventive examinations by general practitioners or dermatologists, the primary presentation of patients with extensive tumors is rare [27]. Most lesions are excised at a small- or medium-sized stage. Yet, if left untreated for years before first consultation, invasive tumor growth with the destruction of underlying soft tissue or bones may occur, as illustrated in Figure 2 and Figure 3 [27]. The frequency of laBCC has been estimated between 1 and 10% compared to 0.0028–0.5% for mBCC [28,29,30].

## 4. Diagnosis

BCC is usually diagnosed clinically with the aid of dermoscopy, which reveals atypical vessels, ovoid nests, and leaf-like structures [24]. Histologically, most cases show nodular or stranded epidermal tumor cell proliferations, originating from the basal layer [31,32]. Histopathology further classifies BCC as nodular (60%), superficial (20%), or least frequently morphea-like [24,27,28].

## 5. Classification/Staging: Defining Locally Advanced and Metastatic BCC

The typical definition of laBCC comprises tumors that are not suitable for standard surgical excision due to size, location (proximity to critical structures), or multiple recurrences [33,34]. LaBCC is often characterized by invasive growth involving deep structures like muscle or bone. In rare cases (about 0.3% of advanced BCCs) even intracranial invasion may occur. The risk of invasive growth depends on the horizontal tumor diameter, number of recurrences, histological subtype, and the presence of other adverse prognostic factors such as perineural invasion or previous radiotherapy (radioderm) [35].

Despite the above-mentioned criteria, a general, clinically applicable, and meaningful definition of laBCC remains challenging. An expert panel consensus thus proposed criteria for laBCC requiring systemic treatment based on the TNM classification and individual patient-specific factors (e.g., recurrence after radiotherapy) [34]. In contrast to other skin tumors (i.e., melanoma), the TNM classification of BCC [36], however, is not routinely used in clinical practice. Therefore, the European Academy of Dermatooncology (EADO) has proposed a novel operational BCC classification separating easy and difficult-to-treat BCC into seven sub-categories (stages I, II A-B, III A-C, IV) [17]. In addition to stage I tumors (low-risk common BCC) and stage IV (metastastic) tumors, five difficult to treat (DTT) categories (IIA-IIIC) were defined. These include common but somewhat DTT BCC owing to their location, recurrence, or poorly defined borders (stage IIA), DTT-BCC due to the multiplicity of tumors (stage IIB), and three different types of laBCC (stage III). These are distinguishable by their extent of resectabilty (IIIA: surgically curable without expected functional mutilations, IIIB: surgically curable with expected functional mutilations, IIIC: not at all surgically curable). Therapeutically, restricted resectability necessitates a multidisciplinary approach including a radio-oncological and/or systemic treatment alternative or adjunct to surgical procedures [17].

Distant metastases from BCC are rare. If so, these most frequently occur in patients with extensive and invasive tumors and can involve regional lymph nodes, lung, bone-marrow, or liver [37,38,39].

The conduction of radiographic imaging is necessary for accurate staging in advanced BCC. Computed tomography should be preferred for the evaluation of bone involvement or distant metastases, while magnetic resonance imaging is typically conducted to evaluate the extent of soft-tissue involvement in the area of the primary tumor or perineural invasion [17].

## 6. Treatment of Primary BCC

Complete surgical excision is the method of choice for common and easy-to-treat BCC, with tumor-free margins being the most important prognostic factor regarding the risk of recurrence [31]. As an alternative to surgery, superficial lesions can be treated with topical agents (5-Fluorouracil, imiquimod or photodynamic therapy) or local ablative procedures such as cryotherapy or electrodessication [27,31,33].

Patients with DTT-BCC who are not suitable for a surgical resection due to advanced age, comorbidities, and/or a potential post-surgical loss of functionality or adverse aesthetical outcome may be considered for primary radiotherapy. Also, adjuvant radiotherapy was shown to improve local tumor control and prevent recurrence, hence being of clinical use in particular therapeutic scenarios with a high risk of recurrence, e.g., after (borderline) incomplete resection of large tumors margins or in the presence of gross perineural involvement [40,41].

## 7. Systemic Treatment of Locally Advanced or Metastatic BCC

Over the last decade, novel therapeutic approaches for patients with laBCC and mBCC not suitable for surgery or radiotherapy have emerged. While earlier treatment regimens mainly involved platinum-based cytotoxic chemotherapy, targeted strategies such as hedgehog inhibitor and immunotherapy, meanwhile, proved to be superior, in terms of both treatment response and tolerability [42,43].

## 8. Vismodegib

The SMO inhibitor vismodegib (Erivedge^®^) was the first hedgehog pathway inhibitor approved for advanced BCC by the FDA and EMA in 2012 and 2013, respectively, and is indicated for patients with laBCC and mBCC not eligible for surgery or radiotherapy [44,45].

ERIVANCE, STEVIE, and MIKIE are the three pivotal phase II trials conducted with vismodegib [24,46,47,48]. The ERIVANCE trial included 104 patients (71 with laBCC not amenable to surgery or radiotherapy, 33 with mBCC) yielding a 60% overall response rate (ORR) in laBCC and 49% in mBCC [49,50]. Comparably, the large-scale STEVIE trial including 1161 patients reported a 69% and 37% ORR in laBCC and mBCC, with complete responses observed in 33% and 5% of patients, respectively [24]. The MIKIE trial investigated two different intermittent dosing regimens of vismodegib in 229 patients with Gorlin syndrome and multiple BCC. The duration of therapy in groups A and B was 12 and 24 weeks, respectively. Both patient cohorts had an 8-week break between treatment cycles. After 73 weeks of treatment, in group A the mean number of BCC was reduced by 62.7% compared to 54.0% in group B [47].

Treatment-related adverse events (AE) were observed in 98 to 100% of patients across all trials, with typical class-specific AEs such as muscle spasms (66–79%), alopecia (62–66%), and dysgeusia (55–67%) occurring most frequently [24,47,48].

Beyond its use in advanced BCC, the VISMONEO study investigated neoadjuvant treatment with vismodegib, aiming to reduce tumor size and minimize the functional and aesthetic impacts of surgery. In this phase II, open-label study, 55 patients with facial BCC were treated with vismodegib for 4 to 10 months (median: 6 months) before surgery. Eighty percent of patients experienced tumor regression, yielding an objective response rate of 71%, with 27 patients (49%) showing a complete response, which was histologically confirmed in 93% of cases. Within 3 years of follow-up, 36% of patients who had initially responded to treatment experienced tumor recurrence [51,52].

## 9. Sonidegib (vs. Vismodegib)

Sonidegib (Odomzo^®^) is the second hedgehog inhibitor available for patients with laBCC unsuitable for surgery or radiotherapy, and received regulatory approval in 2015 based on the results of the BOLT trial [10,26,53].

In 2020, 3.5 years after study completion, the final analysis of this phase II trial with 230 patients was published [54]. ORRs of 56.1% and 8% were reported in patients with laBCC and mBCC, respectively. The disease control rate was over 90% in both laBCC and mBCC. However, no patient with mBCC achieved a complete response. The median duration of response for patients with laBCC was 26.1 months. In both the laBCC and the mBCC cohorts, the median time of treatment exposure was 11 months. Figure 4 summarizes the efficacy of sonidegib and compares it to the results of the pivotal ERIVANCE trial with vismodegib [45].

AEs grade 3 or higher were reported in 43% of patients, leading to AE-related treatment interruption in one third of patients. Similar to vismodegib, the most common AEs were muscle spasms in 54.4% of patients, followed by alopecia and dysgeusia in 49.4% and 44.3% of patients, respectively [55]. Most frequent sonidegib-related AEs in laBCC patients are depicted in Figure 5 and compared to the results of the large safety-trial with vismodegib STEVIE [55]. Closer characterization and management recommendations of these class-specific HHI-related AEs have been provided elsewhere [54].

Overall, the efficacy and safety data of sonidegib in laBCC in the BOLT trial were similar to vismodegib. In contrast, sonidegib was not approved for metastatic disease, since the ORR of sonidegib in mBCC was 23%, which appears lower than the 49 and 37% reported for vismodegib in the ERIVANCE and STEVIE trial, respectively [45,55]. However, the small sample size (*n* = 36) in BOLT’s mBCC group limits direct comparison. Hence, more trials would be needed to reliably assess sonidegib’s efficacy in mBCC [48,54,55]. Odom et al. compared BOLT and ERIVANCE data, showing a slightly higher ORR, longer median progression-free survival (PFS), and duration of response (DOR) with sonidegib [54,57]. In STEVIE, in turn, the ORR (69%) was similar to BOLT (71%), overall indicating similar efficacy of vismodegib and sonidegib [54,55].

In contrast to vismodegib, only the continuous dosing of sonidegib has been specifically evaluated in a small study of ten patients with Gorlin syndrome (sonidegib 400 mg daily in 8 and placebo in 2 patients), achieving complete histological clearance in 57% [45,47,58]. Similarly, only case reports suggest neoadjuvant efficacy of sonidegib. A phase 2 trial with pre-operative sonidegib is currently ongoing (NCT03534947) [59,60].

A prospective Dutch registry study is currently enrolling participants to assess the efficacy and tolerability of HHI for the treatment of laBCC and mBCC, also including patients with Gorlin syndrome. With PFS as its primary outcome measure, the study aims to evaluate real-world HHI efficacy and to directly compare vismodegib and sonidegib (NCT05463757).

## 10. Hedgehog Inhibitor Resistance and Alternative Treatment Strategies

Hedgehog inhibitor therapy can be halted or discontinued for different reasons: either in patients with ongoing complete response [61] or in those experiencing the occurrence of higher-grade adverse events or disease progression indicating either intrinsic or acquired treatment resistance. The mechanisms behind, and the potential biomarkers for, resistance are still under investigation [62,63]. In 2016, Soura et al. proposed hair regrowth as a potential clinical indicator of HHI resistance in advanced BCC [64]. The development of treatment resistance occurs largely through mutations of SMO, either altering its drug-binding site or conferring constitutive SMO activity. Also, the activation of GLI through non-canonical Hedgehog pathway activation independent of PTCH1-SMO interaction can reduce SMO inhibitor effectiveness [64,65,66,67]. Switching between HHIs has been attempted to overcome resistance without convincing evidence. In 2016, Danial et al. found no added benefit of sonidegib after vismodegib failure in nine advanced BCC patients [68]. In contrast, a 2017 case by Yoon et al. reported tumor regression with sonidegib and itraconazole, which also inhibits SMO through a distinct mechanism after vismodegib failure [69].

Second-generation HHI have been designed to prevent the occurrence of SMO-mediated HHI resistance. These small molecules, such as the Casein Kinase 2 (CK2) inhibitor silmitasertib (CX-4945, NCT03897036), inhibit terminal components of the SHH pathway. The results of ongoing phase I and II trials are yet to be published.

In addition to systemic HHI therapy, the topical application of the SMO inhibitor patidegib could represent a possible treatment alternative, particularly for patients with Gorlin Syndrome and multiple BCC. Recently published results of a phase II trial suggest that the application of patidegib gel may reduce the incidence of new BCC. In this study, 17 patients with a total of 85 surgically eligible BCCs (SEBs) were randomized 1:1:1 to receive treatment with either vehicle control, or patidegib topical gel in a two or four percent concentration twice daily for 26 weeks.

The primary endpoints were defined clinically as the percentage decrease in the largest diameter of treated SEBs after 26 weeks of treatment and on a molecular level as changes in glioma-associated oncogene homolog 1 (GLI1) messenger RNA (mRNA) expression in tumors treated with either patidegib gel or vehicle after 6 weeks of treatment.

Secondary endpoints included (a) the number of new SEBs (defined as facial BCCs with a diameter ≥ 5 mm) that were not detected at baseline and (b) the proportion of facial BCCs (excluding the nose and periorbital area) with the largest diameter of less than 5 mm at baseline and/or week 2 that had grown to a diameter of more than 5 mm at predefined visits to the study center.

Although the primary endpoints of the study were not met, post-hoc analyses showed that patients treated with patidegib developed significantly fewer new facial SEBs compared to vehicle (0.3 vs. 1.4, respectively; *p* = 0.008).

Further post-hoc per-protocol (PP) analyses yielded a more than double reduction in diameter in patidegib treated tumors (mean decrease 51% vs. 22%; *p* = 0.038) and significantly more complete responses (12/48 vs. 0/16, *p* = 0.021) compared to vehicle [58].

Based on these phase II findings, patidegib was granted Orphan Drug Designation by the EMA and Breakthrough Therapy Designation by the FDA in 2018 and subsequently evaluated in a double-blinded phase III trial including 174 Gorlin syndrome patients with at least ten clinically typical BCC who received either patidegib 2% gel or vehicle (NCT03703310) for 12 months. The primary endpoint was the number of new BCC developing during this period. The initial results (available at https://clinicaltrials.gov) show that patients treated with topical patidegib developed a mean of 2.84 new BCC during 12 months compared to 4.03 BCC in patients receiving topical vehicle. Also, there was an almost 10% reduction in the incidence of new basal cell carcinomas on the face (54.2 vs. 64% in patidegib and vehicle patients, respectively). The detailed statistical analyses of this study are due to be published [70].

To further explore the last finding mentioned above, another ongoing double-blind phase III trial specifically compares the number of new BCCs developing on the face of participating patients with Gorlin syndrome after 12 months of topical treatment with either with patidegib 2% gel or vehicle (NCT06050122).

Programmed death-1 (PD-1) antibodies and other immune-checkpoint inhibitors proved to be highly effective in the treatment of melanoma and other forms of skin cancer such as cutaneous squamous cell or Merkel cell carcinoma. As BCC exhibits the highest mutational burden of all cancers, immune-checkpoint inhibitor therapy was expected to potentially become a valuable therapeutic option for patients with advanced disease. This assumed positive effect of a high tumor mutational burden on immune-checkpoint inhibitor efficacy may be mitigated by a reduced immunogenicity owing to impaired antigen presentation, diminished T-cell infiltration and increased levels of immunosuppressive cytokines such as interleukin 10 found in the tumor microenvironment of BCC [71].

Against this background, a pivotal phase II trial eventually demonstrated that the PD-1 inhibitor cemiplimab (Libtayo^®^) was, indeed, effective in patients with locally advanced BCC, who were unsuitable for or had recurrence after HHI treatment. Out of 84 patients, 31% experienced an objective response, with a median time to response of 4.3 months and a disease control rate of 60% [72]. Hence, cemiplimab was approved for the second-line treatment of advanced BCC in February 2021. In the extended analysis of this study in 2024, after a median duration of follow-up of 15.9 months, the ORR slightly increased to 32.1% per independent central review [73]. Cemiplimab showed comparable activity in 54 mBCC patients previously treated with HHI with an ORR of 22% and a DCR of 63% [74].

In order to evaluate the tolerability and efficacy of intralesional administration of cemiplimab in patients with BCC and SCC, a phase I clinical trial is currently recruiting patients. During the initial phase of this study, different doses of cemiplimab are administered weekly for 12 weeks (NCT03889912).

Other PD1-antibodies and PD-1-based checkpoint inhibitor combinations are under investigation. The ongoing clinical trial NCT03521830 evaluates nivolumab (anti-PD-1) monotherapy in treatment-naïve and HHI-experienced patients, as well as nivolumab combined with relatlimab (anti-LAG-3) or nivolumab with ipilimumab (anti-CTLA-4) in patients with anti-PD-1-refractory advanced BCC. Preliminary results on 22 evaluable patients have been presented showing a promising 50% response rate in 10 treatment-naive patients who received nivolumab monotherapy [75]. Already in 2019, the PD-1 antibody pembrolizumab (Keytruda^®^) alone or with vismodegib was tested in 16 advanced BCC patients. Pembrolizumab monotherapy yielded an ORR of 44%, compared to 29% in patients receiving combination therapy, resulting in an overall ORR of 38% at 18 weeks [76]. Another ongoing trial investigating combined anti-PD-1- and HHI-treatment assesses the efficacy of cemiplimab in combination with pulsed (2-weeks-on/2-weeks-off) sonidegib therapy in 20 advanced BCC patients (NCT04679480). Trial completion is estimated at the end of 2024.

In case of failure of both HHI and immune-checkpoint inhibitor treatment, current guidelines recommend electrochemotherapy in patients with tumors suitable for this local procedure [77]. Response rates of up to 96% have been reported for typically bleomycin-based electrochemotherapy [49,78], which, also in our experience, represents a well-tolerated treatment option in selected patients.

Regarding neoadjuvant BCC treatment, intralesional oncolytic virotherapy with talimogene laherparepvec showed promising results in an exploratory phase II study in 18 patients with a difficult to resect BCC [78]. After 6 cycles of intralesional treatment (13 weeks) before surgery, the primary endpoint avoidance of skin graft or skin flap surgery was reached in 53% of patients. Partial and complete tumor responses were observed in 24 and 35% of patients, respectively. Other intralesional treatments under investigation include the oncolytic virus RP1, the immune modulator daromun as well as intralesional cemiplimab. Also, neoadjuvant pembrolizumab monotherapy is currently investigated in a phase I trial in patients with locoregionally advanced, but resectable BCC of the head and neck (NCT04323202).

A Phase 2 clinical trial has been designed to investigate the combination of the lymphocyte-activation gene 3 (LAG 3) inhibitor relatlimab and the PD-1 inhibitor nivolumab versus nivolumab (2:1 randomization) in a neoadjuvant setting in patients with resectable high-risk BCC. Patients will receive 4 cycles of therapy every 4 weeks, followed by surgical resection. The primary outcome pathologic response rate will be evaluated at resection, with monitoring for recurrence and secondary outcomes thereafter (NCT06624475).

An overview of above-mentioned clinical trials is provided in Table 1.

Clinically, basosquamous carcinomas (BSC), sometimes referred to as “metatypical carcinomas”, are barely distinguishable from BCC. Histologically, they represent a controversially discussed separate entity, sharing characteristics of both BCC and SCC [79]. BSC have been proposed to derive from BCC with genetic alterations leading to squamous differentiation [80,81]. Regarding their potential to develop metastases, the National Comprehensive Cancer Network (NCCN) describes BSC as resembling SCC rather than BCC [82,83]. Prospective controlled trials investigating systemic treatment with HHI or anti-PD(L)1 antibodies therapy in advanced BSC have not been conducted [84]. Despite several case reports suggesting activity of HHI, their use in BSC is controversial, as HHI treatment may promote SCC development, which could particularly be relevant regarding the SCC component of BSCs. A recent systematic review identified 23 patients developing same-site cutaneous malignancies during HHI treatment of laBCC, including six (26%) BSCs. An exact mechanism by which inhibition of the SHH pathway could promote SCC development remains to be identified [85,86,87,88,89,90].

## 11. Summary and Conclusions

The treatment landscape of advanced BCC has undergone significant transformation in recent years. In advanced disease not amenable to surgery or radiotherapy, the similarly efficacious HHI vismodegib and sonidegib currently remain standard-of-care for first-line treatment. Yet, primary or acquired treatment resistance, as well as class-specific adverse events, limit the long-term feasibility and efficacy of HHI therapy. Immune-checkpoint inhibitors such as cemiplimab have taken on an important role in the resulting second-line treatment scenario, while large-scale controlled first-line studies with PD-1 inhibitors remain to be conducted. Resembling other types of skin cancer, current research focuses on combined targeted and immunotherapy, neoadjuvant treatment approaches, as well as potentially less toxic intralesional and topical treatments. Based on a novel, clinically applicable BCC classification, and ideally combined with reliable biomarkers, these treatment strategies may further advance care and improve the prognosis of our patients with locally advanced or metastatic BCC in the near future.

## Figures and Tables

**Figure 1 cancers-17-00068-f001:**
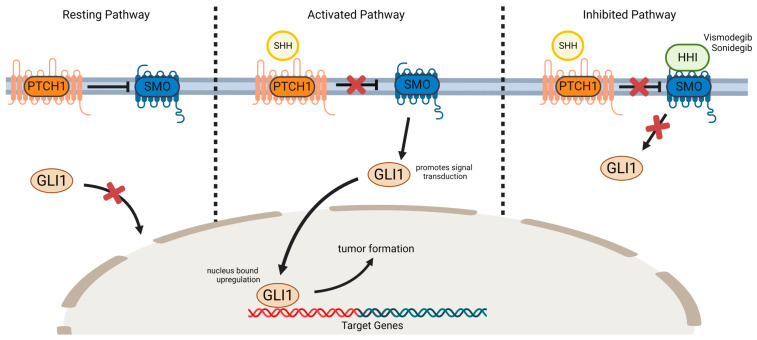
Hedgehog signaling pathway. In the resting state, PTCH1 inhibits SMO, which in turn prohibits the activation of GLI. Upon binding SHH with PTCH1, the surface-membrane protein SMO is disinhibited, resulting in the nucleus-bound upregulation of GLI1 and, consecutively, tumor formation. In the case of HHI binding to SMO, GLI1 upregulation is disengaged. Abbreviations: GLI1: glioma associated oncogene 1; HHI: hedgehog inhibitor; PTCH1: Patched; SMO: smoothened; SHH: sonic hedgehog.

**Figure 2 cancers-17-00068-f002:**
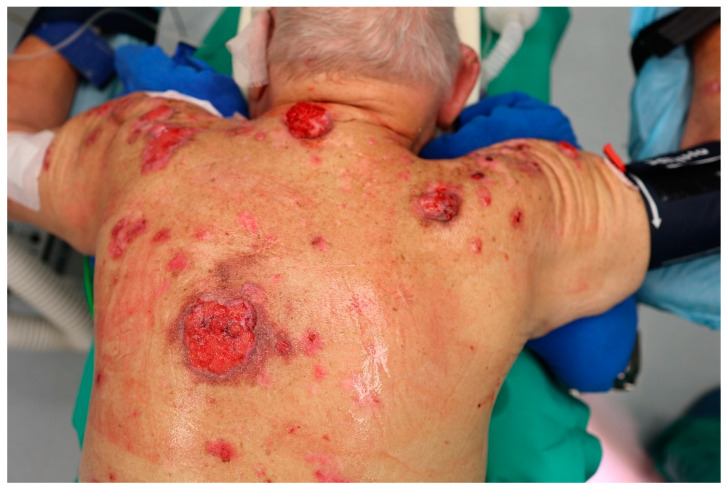
Clinical presentation of multiple advanced BCC in a patient with Gorlin syndrome (source: Department of Dermatology and Allergology, Paracelsus Medical University, Salzburg, Austria).

**Figure 3 cancers-17-00068-f003:**
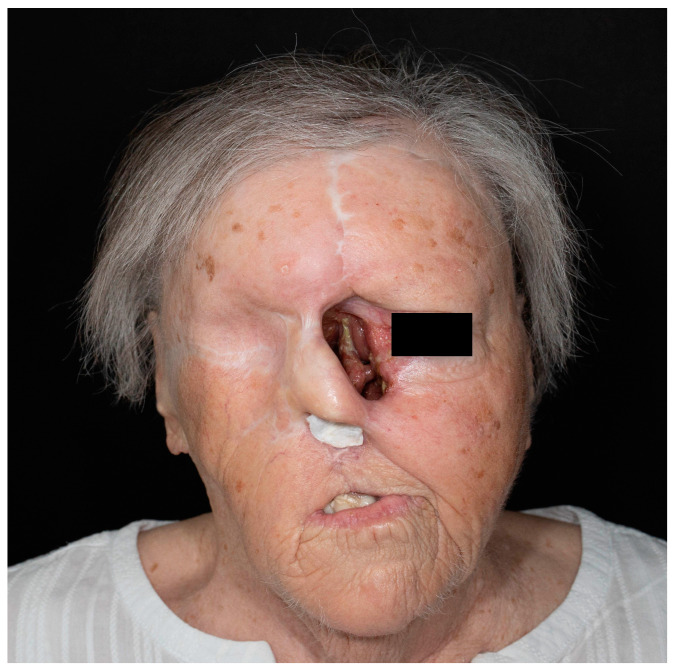
Clinical presentation of locally advanced BCC exhibiting extensive destruction of nasal cartilage and orbital bones (source: Department of Dermatology and Allergology, Paracelsus Medical University, Salzburg, Austria).

**Figure 4 cancers-17-00068-f004:**
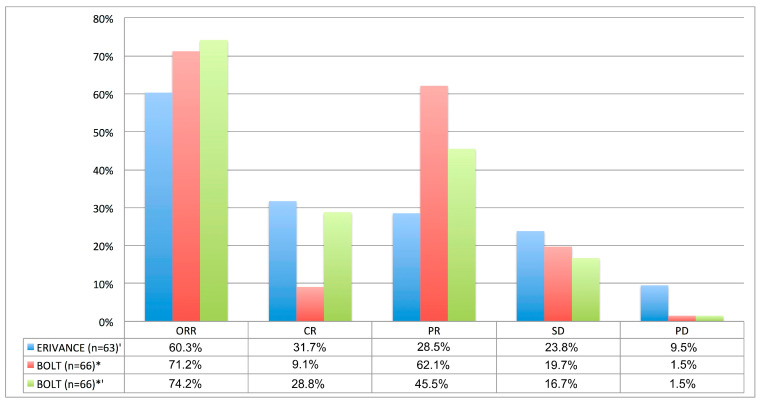
Treatment Response (assessed by investigator-review) in patients with laBCC treated with vismodegib/sonidegib in the ERIVANCE`/BOLT+ trial [26,45,46]; ′ 39-month analysis, adjusted with RECIST criteria; * 42-months analysis, adjusted with mRECIST criteria; *′ 42-month analysis adjusted with ERIVANCE-like RECIST criteria. Abbreviations: CR, complete response; ORR, objective response rate; PD, progressive disease; PR, partial response; SD, stable disease.

**Figure 5 cancers-17-00068-f005:**
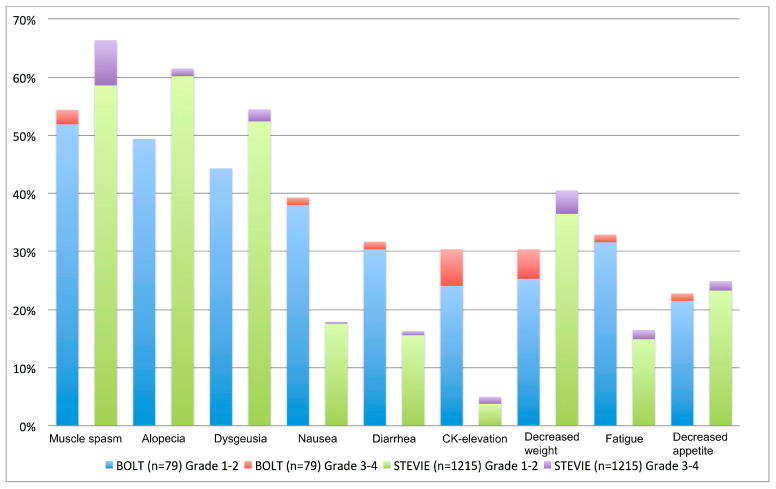
Comparison of most frequent treatment-related adverse events of Sonidegib in the BOLT and of Vismodegib in the STEVIE trial [55,56].

**Table 1 cancers-17-00068-t001:** Overview of ongoing clinical trials in locally advanced or difficult to treat BCC11. Therapeutic approach in case of basosquamous carcinoma.

NCT	Title of the Study	Phase	Study Drug	Number of Patients (Recruitment Goal)	Status
NCT06050122	Efficacy and Safety of Patidegib Gel 2% for Preventing Basal Cell Carcinomas on the Face of Adults with Gorlin Syndrome	III	Patidegib Topical Gel Patidegib Topical Gel with no active patidegib	140	Recruiting
NCT03703310	Study of Patidegib Topical Gel, 2%, for the Reduction of Disease Burden of Persistently Developing Basal Cell Carcinomas (BCCs) in Subjects with Basal Cell Nevus Syndrome (Gorlin Syndrome)	III	Patidegib Topical Gel, 2% Patidegib Topical Gel, Vehicle	174	Completed
NCT03889912	Intralesional Cemiplimab for Adult Patients with Cutaneous Squamous Cell Carcinoma or Basal Cell Carcinoma	I	Cemiplimab	113	Recruiting
NCT04679480	Anti-PD1-antibody and Pulsed HHI for Advanced BCC	II	Cemiplimab	20	Active Not recruiting
NCT03521830	Nivolumab Alone or Plus Relatlimab or Ipilimumab for Patients with Locally-Advanced Unresectable or Metastatic Basal Cell Carcinoma	II	Nivolumab Ipilimumab Relatlimab	57	Recruiting
NCT04323202	Neoadjuvant-Adjuvant Pembrolizumab in Resectable Advanced Basal Cell Carcinoma of H&N	I	Pembrolizumab	13	Active Not recruiting
NCT06624475	Neoadjuvant Opdualag Versus Nivolumab for Resectable High-Risk Basal Cell Carcinoma	II	Nivolumab Opdualag	30	Not yet recruiting
NCT03534947	A Study to Evaluate Neoadjuvant Sonidegib Followed by Surgery or Imiquimod in the Management of Basal Cell Carcinoma (SONIB)	II	Sonidegib Imiquimod	20	Recruiting
NCT05463757	Oral Hedgehog Inhibitors in the Treatment of Basal Cell Carcinoma in the Netherlands: a Prospective Registration Study	Observational	Vismodegib Sonidegib	80	Recruiting
NCT03897036	Treatment Duration Increment and Pharmacodynamic Study of CX-4945 in Patients with Basal Cell Carcinoma (BCC)		I CX-4945	25	Active, not recruiting

## Data Availability

No new data were created in this study. Data sharing is not applicable to this article.

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
