# Peer review of "Therapeutic Approaches for Advanced Basal Cell Carcinoma: A Comprehensive Review"

_cancers, 2024, doi:10.3390/cancers17010068_

Round 1
Reviewer 1 Report
Comments and Suggestions for Authors
In the current manuscript in Cancers, specialty section Cancer Therapy (Manuscript ID: cancers-3324800), Magdalena Hoellwerth et al. have explored on the therapeutic approaches for advanced basal cell carcinoma. With an emphasis on both locally progressed and metastatic basal cell carcinoma disease, this review study looks at and summarizes the therapeutic methods for advanced basal cell carcinoma.
Overall, this is a well-compiled study with important insights that focuses on the broad summary of advanced basal cell carcinoma management and lists authorized systemic treatments, including as immunotherapy and hedgehog inhibitors. The review additionally investigates at current studies on new approaches to treatment.
In general, this publication advances the knowledge about basal cell carcinoma and its approved possible therapies for improving the overall evolving treatment plans. This manuscript could attract substantial interest from readers of Cancers with specialized section of cancer therapy.
However, I do recommend minor modifications, and addressing these could further strengthen the manuscript:
1. Line 80: “Tumor size and vertical extension vary widely.” This sentence does not convey anything. Please rewrite it to make sense in reference to the previous statements.
2. Line 87: What is destructive tumor??? Please reframe it with appropriate choice of terms in context of tumors.
3. Line 88: “Yet, if left untreated 88 for years before first consultation, large tumor expansion may occur”. Reframe it.
4. Section 4, Diagnosis: Author should discuss more in reference to the diagnosis of the BCC.
5. Figure 2: Provide the source of the image and how does it represent the clinical presentation of laBCC. Authors have not discussed the image in the text.
6. Line 105: “Risk factors for invasiveness include large size…”
Comment: What do you mean by large size? Large size of what?
Rewrite the whole sentence in more meaningful way.
7. Line 122-123: “In the rare case of distant BCC metastases, these most frequently occur in patients 122 with large tumors and can involve regional lymph nodes, lung, bone-marrow or liver.” Reframe it to make it meaningful sentence.
8. Line 125: “Radiographic staging in advanced BCC depends on the site and suspected extent of 125 the disease.”
Comment: This sentence is not clear. Reframe it. The radiology can be an effective tool in determining the stage of BCC and depending on the location in the metastasis, the technique will be preferred.
9. Section 5, Classification: I understand that the review focus on the therapeutic approaches for advanced BCC. But if you are talking about the classification, you should do it properly. You have not talked about stage IIA and IIB.
10. Line 146: “platin-based chemotherapy” What is platin based therapy???
11. Line 157 – 158: “ the large-scale STEVIE trial including 1.161 patients reported…” I think it is 1,161 patients. Please correct the typo.
12. Line 218: “Hedgehog inhibitor therapy is typically discontinued due to complete response…”. Is it complete response or incomplete response??? I am not sure what author means.
13. Line 221-223: “Second-generation HHI such as silmitasertib (CX-4945, NCT03897036) aim at overcoming SMO-mediated HHI resistance through inhibition of the SHH pathway further downstream.”. Reframe it.
14. Line 243-245: “Programmed death-1 (PD-1) antibodies and other immune-checkpoint inhibitors have been proven to be highly effective in the treatment of melanoma and other forms of skin 244 cancer such as cutaneous squamous cell or Merkel cell carcinoma.” Add been to make it correct.
15. Line 245-251: “Given the fact that BCC 245 exhibit the highest mutational burden of all cancers [105], on the one hand, immune checkpoint inhibitor therapy was expected to potentially become a valuable therapeutic option for patients with advanced BCC. On the other hand, compared to cutaneous squamous cell carcinoma, reduced immunogenicity was described in BCC, owing to impaired antigen presentation, diminished T-cell infiltration and increased levels of immunosuppressive cytokines such as interleukin 10 in the tumor micro-environment. [69]”. Rewrite it to make more sense. It is not properly clear to the reader what author wants to say.
16. A pictorial or tabular representation of therapeutic approaches defining the timelines and drugs for BCC, will substantially add to the review and make it more effective for the readers.
Comments on the Quality of English Language1. The English needs a slight improvement as small grammatical mistakes are present throughout. Please carefully scrutinize it.
2. Some sentences are too small and can be combined to make an effective reading.
Author Response
We thank Reviewer 1 for the constructive feedback.
We have adapted the manuscript accordingly, as outlined in the attachment.

Reviewer 2 Report
Comments and Suggestions for Authors
The review by Hoellwerth et al. is complete and well-written, but it lacks of novelty, since many other similar articles are available on the same topic.
I would suggest to consider reverting the paper and focusing on the novelties in the treatment of laBCC (i.e., Patidegib and further) that, in the present version, represent only a small part in the final sentences in the Discussion.
Author Response
We thank Reviewer 2 for the feedback and for acknowledging the quality of our manuscript. We appreciate the suggestions and the opportunity to further refine our work.
The adaptions are outlined in the attachment.

Reviewer 3 Report
Comments and Suggestions for Authors
The authors performed a study about Therapeutic approaches for advanced Basal Cell Carcinoma. The article is of interest however some changes are needed:
- I suggest to add more clinical pictures of localy advanced BCC
- Please add a paragraph about common side effects of Hedgehog inhibitors
- Please add some relevant sentence about metastatic BCC
- Please explain also the therapeutic approach in case of basosquamous carcinoma
- Regarding the pathogenesis, as reported for melanoma and bcc, in the discussion, please add also some relevant speculation about the role of vitamin d and anatomic stites of the primary tumors as reported in these two interesting studies that you can add in the references: The Relationship between Vitamin D and Basal Cell Carcinoma: A Systematic Review. PMID: 36312675 AND Clinicopathological features, vitamin D serological levels and prognosis in cutaneous melanoma of shield-sites: an update. Med Oncol. 2015 Jan;32(1):451. PMID: 25516505.
Thank you
Author Response
We thank Reviewer 3 for the insightful comments and constructive suggestions.
Our point-by-point response is provided in the attachment.

Round 2
Reviewer 2 Report
Comments and Suggestions for Authors
The article has been significantly improved and the Authors have addressed the required issues.
Reviewer 3 Report
Comments and Suggestions for Authors
The authors improved the manuscript and can be accepted for publication
Thank you